# Safety and Efficacy in the Transcortical and Transsylvian Approach in Insular High-Grade Gliomas: A Comparative Series of 58 Patients

**DOI:** 10.3390/curroncol32020098

**Published:** 2025-02-10

**Authors:** Alberto Morello, Francesca Rizzo, Andrea Gatto, Flavio Panico, Andrea Bianconi, Giulia Chiari, Daniele Armocida, Stefania Greco Crasto, Antonio Melcarne, Francesco Zenga, Roberta Rudà, Giovanni Morana, Diego Garbossa, Fabio Cofano

**Affiliations:** 1Neurosurgery Unit, Department of Neuroscience “Rita Levi Montalcini”, “Città Della Salute e Della Scienza” University Hospital, University of Turin, 10126 Turin, Italy; fr.rizzo@unito.it (F.R.); andrea.gatto@unito.it (A.G.); flavio.panico@unito.it (F.P.); anmelcarne@gmail.com (A.M.);; 2Department of Neurosurgery, Ospedale Policlinico San Martino, Istituto di Ricovero e Cura a Carattere Scientifico (IRCCS) for Oncology and Neuroscience, 16132 Genoa, Italy; andrea.bianconi@edu.unito.it; 3BCAM Bilbao Center for Applied Mathematics—Mazarredo Zumarkalea, 48009 Bilbao, Spain; gchiari@bcamath.org; 4A.U.O. “Policlinico Umberto I”, Neurosurgery Division, Human Neurosciences Department, Sapienza University, 00185 Rome, Italy; daniele.armocida@unito.it; 5Magnetic Resonance Imaging Service, R.I.B.A. and Larc Centers, 10128 Turin, Italy; sgrecocrasto67@gmail.com; 6Skull Base and Pituitary Surgery Unit, “Città della Salute e della Scienza” University Hospital, 10126 Turin, Italy; 7Division of Neuro-Oncology, Department of Neuroscience “Rita Levi Montalcini”, “Città Della Salute e Della Scienza” University Hospital, University of Turin, 10126 Turin, Italy; 8Division of Neuroradiology, Department of Diagnostic Imaging and Radiotherapy, “Città Della Salute e Della Scienza” University Hospital, University of Turin, 10126 Turin, Italy

**Keywords:** glioma, glioblastoma, insula, insular glioma, oncology

## Abstract

Gliomas within the insular region represent one of the most challenging problems in neurosurgical oncology. There are two main surgical approaches to address the complex vascular network and functional areas around the insula: the transsylvian approach and the transcortical approach. In the literature, there is not a clear consensus on the best approach in terms of safety and efficacy. The purpose of this study is to evaluate the effectiveness of these approaches and to analyze prognostic factors on the natural history of insular gliomas. Patients with newly diagnosed high-grade insular gliomas who underwent surgery between January 2019 and June 2024 were analyzed. The series was analyzed according to the classification of Berger–Sanai and Yaşargil. The Karnofsky performance score (KPS), extent of resection (EOR), progression-free survival (PFS), and overall survival (OS) were considered the outcome measures. A total of 58 primary high-grade insular glioma patients were enrolled in this study. The IDH mutation was found in 13/58 (22.4%); specifically, 3/13 (23.1%) were grade 4, and 10/13 (76.9%) were grade 3. Furthermore, 40/58 patients (69%) underwent gross total resection (GTR), 15 patients (26%) subtotal resection, and 3 patients (5%) partial resection. Middle cerebral artery encasement negatively affected the OS. GTR, radiotherapy, KPS, and autonomous deambulation at a month after surgery positively affected the OS. The surgical approach used was transsylvian and transcortical in 11 and 47 cases, respectively. The comparison between the two different approaches did not display differences in terms of neurological deficits and OS (*p* > 0.05). The transcortical approach was related to the greater achievement of GTR (*p* = 0.031). According to the Berger–Sanai classification, the transcortical approach has higher EOR and postoperative KPS when the lesion is in zone III-IV (*p* = 0.029). Greater resection of insular gliomas can be achieved with an acceptable morbidity profile and is predictive of improved OS. Both the transsylvian and transcortical corridors to the insula are associated with low morbidity profiles. The transcortical approach with intraoperative mapping is more favorable for achieving greater EOR, particularly in gliomas within the inferior border of the Sylvian fissure.

## 1. Introduction

Insular gliomas (INGs) continue to pose significant challenges in neurosurgical oncology, despite advancements in their treatment. The complexity of these tumors stems from their intricate vascular structure, involving the middle cerebral artery (MCA) and lenticulostriatal arteries (LTSas), as well as the functional organization of both cortical and white matter regions [1]. Furthermore, the insula plays a crucial role in various cognitive functions, including emotional processing, memory, and social capabilities [2]. The current standard of care for INGs involves maximizing safe tumor resection, followed by a combination of chemotherapy and radiation therapy [3]. According to the published studies, the incidence of temporary postsurgical deficits ranges from 14% to 59%, while permanent deficits occur in 0% to 20% of cases.

There is no general consensus on the choice of the appropriate surgical approach. Several studies have been reported in the literature regarding this, especially on the two main corridors: transcortical and transsylvian.

In the transsylvian approach, a wide dissection of the Sylvian fissure is typically performed to achieve sufficient exposure of the insular region. This procedure can be challenging when the Sylvian cistern ends posteriorly. However, a significant benefit of the transsylvian technique is that it spares the frontal and temporal opercula in the dominant hemisphere, thereby reducing the risk of direct damage to the language network [4]. With advancements in cortical stimulation and functional brain mapping, the transcortical approach to accessing the insula is also becoming more widely used. For large gliomas, multiple cortical windows are created through non-functional areas of the cortex and are connected at the resection-cavity level, preserving both functional cortical areas and critical Sylvian vessels [5].

In the literature, there is no general consensus on the choice of the appropriate surgical approach. The choice of one technique over the other remains subjective and dependent on the surgeon’s experience. The aim of this study is to evaluate the effectiveness and the safety of these approaches and to analyze prognostic factors on the natural history of insular gliomas.

## 2. Materials and Methods

### 2.1. Patient Selection

We analyzed 58 consecutive patients with high-grade insular glioma (grade 3–4) who underwent resection using the transcortical or transsylvian approach, in the University Hospital of Turin “Città della Salute e della Scienza”, from January 2019 to June 2024. All the surgeries were performed by the senior authors (A.M. and D.G.). Histological inclusion criteria were chosen based on the WHO classifications of 2016 and 2021. All adult patients affected by newly diagnosed high-grade gliomas were included in this study, while those with low-grade gliomas, infratentorial tumor location, and multicentric lesions were excluded. The preoperative and postoperative clinical data and the adjuvant therapy’s effects on patient survival were analyzed. Informed consent was obtained from all the individual participants included in this study. This study was conducted in compliance with Good Clinical Practice guidelines and according to the Helsinki declaration for ethical human rights and ethical standards of the authors’ institutions.

### 2.2. Tumor MR Analysis

The tumors were classified into various zones based on the Berger–Sinai and Yaşargil classification using MR images [6,7].

According to the Berger–Sanai classification (BSC), the insula is divided into four distinct zones. The anterior–posterior boundary is determined by a line passing through the foramen of Monro, while the superior–inferior boundary is defined by the Sylvian fissure. This classification system facilitates the assessment of insular gliomas in relation to their functional anatomy, including the peri-Sylvian language network (zones I–III), the primary motor and sensory regions (one II), Heschl’s gyrus (zone III), and the deep lenticulostriate arteries (zone IV). Gliomas that involve all 4 zones are referred to as “Giant” insular gliomas.

In contrast, the Yasargil classification (YC) categorizes tumors based on their extension as follows: type 3A refers to tumors confined solely to the insula; type 3B involves tumors that extend into the adjacent operculum; type 5A describes tumors that extend into the frontoorbital and/or temporopolar regions without affecting the mesiotemporal structures; and type 5B includes tumors that extend into the frontoorbital and temporopolar regions with the involvement of the mesiotemporal structures.

The MR images were reviewed by one of the expert glioma neuroradiologists (S.C.) for the tumor volume (volume of contrast-enhancing tissue on gadolinium-contrasted T1-weighted images), putamen and ependymal involvement, and the extent of tumor resection (EOR). All the postoperative scans were completed within 72 hours of the surgery. The EOR was calculated as follows, following the resection classification of the RANO group [8]: (preoperative-postoperative tumor volume)/preoperative tumor volume. Subsequently, to simplify the analysis, the EOR was subdivided into gross total (>90%) (GTR), subtotal (70–90%) (STR), or partial (<70%) resection.

### 2.3. Surgical Technique

The choice of one technique over the other, between transcortical or transsylvian, depends on the preference of senior surgeons, based mainly on the extent of the lesion and its relationship with arterial vessels. The range of cortical resection depends on the region with the gliomas, neurophysiological monitoring (including cortical and subcortical mapping) and preoperative BOLD (blood oxygen level-dependent) fMRI, and diffusion tensor imaging (DTI) tractography information.

Meanwhile, the surgical time is defined as the total “skin-to-skin” operating time (from incision until the final suture is completed).

### 2.4. Statistical Analysis

We adopted the Pearson correlation coefficient as a statistical measure to quantify the degree of the linear relationship between different clinical data. We obtained a coefficient between −1 and +1, where +1 indicates a perfect positive linear correlation, −1 indicates a perfect negative linear correlation, and 0 indicates no linear relationship. The significance test associated with Pearson’s correlation coefficient (resulting in the *p*-value outcome) assesses whether the observed correlation is statistically significant. Statistical significance was set at *p* < 0.05. Overall survival (OS) was defined as the interval from the date of histological diagnosis to date of death or last follow-up. Progression-free survival (PFS) was the interval from histological diagnosis to the first radiological evidence of disease progression (on MRI) or the date of last follow-up or death, whichever was earlier. The Kaplan–Meier method was used to estimate the OS. The log-rank test, also known as the Peto–Mantel–Haenszel test, was used to evaluate the null hypothesis of no difference in survival between the two groups. The analyses were computationally carried out using Python (“scipy.stats” library) (version 3.11.3).

## 3. Results

### 3.1. Patient Demographics

A cohort of 58 patients (24 females (41.4%) and 34 males (58.6%)) was involved in this study. At the date of surgery, the median age was 57.4 years (range 24–80). A median of 90 [range 40–100] and 90 [range 50–100] was obtained for the Karnofsky performance score (KPS) at diagnosis and at 1-month follow-up, respectively (Table 1).

A total of 13 WHO grade 3 (22.4%) and 45 WHO grade 4 (77.6%) insular gliomas were collected. The IDH mutation was found in 13/58 (22.41%); specifically, 3/13 (23.10%) were grade 4, and 10/13 (76.90%) were grade 3. The average percentage value of ki-67 was 35 (2–80).

Overall, 33 tumors were located in the right hemisphere (56.9%) and 25 in the left (43.1%). The median preop enhancement tumor volume was 27 cm^3^ (range 1.7–94.4 cm^3^). The median FLAIR volume tumor volume was 64 cm^3^ (range 1–144.5 cm^3^).

According to the Berger–Sanai insular glioma classification, 33.4% were “Giant” tumors, 22.2% in zone III–IV, 18.5% in zone II–III, 7.4% in zone I + II, 7.4% in zone II, 7.4% in zone I-IV, and 3.7% in zone III.

According to the YC, 35.7% of the tumors were type 3B, 35.7% type 5B, 14.3% type 3A, and 14.3% extended into the frontoorbital and/or temporopolar areas without the involvement of mesiotemporal structures (type 5A). It was possible to analyze the two variables, BSC and YC, in 28 patients.

### 3.2. Surgical Resection

Overall, 40 patients (69%) underwent GTR, 15 patients (26%) underwent STR, and 3 patients (5%) underwent partial resection. According to the BSC, the GTR rates of different types of insular gliomas were as follows: zone I–II (0%), zone II (100%), zone II-III (60%), zone III (10.0%), zone I-IV (100.0%), zone III–IV (83%), zone I–II–III–IV (100%), Giant (50%). According to the YC, the GTR rates were as follows: 3A (50%), 3B (50%), 5A (75%), 5B (90%). The surgical approach used was transsylvian and transcortical in 11 and 47 cases, respectively. Of all the patients, only three cases underwent awake surgery (5%).

### 3.3. Follow-Up and Outcome

There were no deaths in the perioperative period. After surgery, 44 patients (75.9%) received chemotherapy, 45 (77.6%) received radiation, 44 patients (75.9%) received radiation plus chemotherapy. Moreover, 13 patients (22.4%) received a craniotomy alone. Neurological complications were divided into short-term (within the first 7 postoperative days) and long-term (1 and 3 months after surgery) complications (Table 2). PFS averaged 9.8 months in the total population, in particular, 10.1 months in the transcortical group and 10.6 months in transsylvian group. OS averaged 18.5 months and ranged from 17.1 months in the transsylvian group to 19.1 months in the transcortical group. At the most recent update, 8 patients were alive and 50 were reported deceased. Two patients were lost to follow-up; however, it was possible to determine from the regional medical records whether they were alive or deceased. The follow-up duration ranged from 1 to 71 months, with a median of 13 months and a mean of 19.13 months.

A poor Charlson comorbidity index (CCI) and MCA encasement negatively affected the OS (Table 3A). GTR, radiotherapy, KPS, and autonomous deambulation at a month after surgery positively affected the OS. The comparison between the two different approaches did not display differences in terms of neurological deficits and OS (*p* > 0.05) (Table 3B). The transcortical approach is related to the greater achievement of GTR (*p* = 0.031). The probability of using the transsylvian approach was higher in cases with MCA encasement (*p* = 0.001). The probability of using the transcortical approach was higher in cases with BSC and YC ≥ 3 (*p* = 0.002 and 0.005, respectively).

According to the BSC, the transcortical approach has a higher EOR and postoperative KPS when the lesion is in zone III–IV (*p* = 0.029). According to the YC, there were no statistically significant associations with pre-, intra- and postoperative variables between the two surgical approaches (*p* > 0.05).

A survival analysis (Kaplan–Meier estimator) was conducted, stratified by EOR (Figure 1), by YC and BSC (Figure 2), and by histological grade (Figure 3). Patients with grade I or II of BSC had a higher risk of death. Patients with 5A and 5B grade of YC had a higher risk of death. However, these results were not statistically significant (*p* = 0.118 and 0.702, respectively). As expected, the Kaplan–Meier curve, based on histological grade, demonstrated worse OS of patients with grade 4 (*p* = 0.048).

## 4. Discussion

In surgical practice, it is common to find glial lesions at the insular level, since the insula seems to be a preferred site for low-grade gliomas; as reported by Duffau et al., about 11% of high-grade supratentorial gliomas (HGGs) and 25% of low-grade gliomas (LGGs) are located in the insular lobe, although recent studies have shown up to 40% of insular lesions to be high-grade gliomas [9,10,11]. Such lesions were considered irresectable until the 1990s, when Professor Yasargil first published a paper describing the transsylvian approach to 240 tumors of the limbic and paralimbic systems [7]. In some centers, patients with deep tumors involving the insula do not undergo surgery but undergo laser interstitial thermotherapy (LITT) to reduce postoperative risks [12]. However, the results in terms of the outcome and OS are yet to be evaluated in detail. The concept of “no-man’s land” is due to the complex anatomical organization, particularly its close proximity to the internal cerebral artery, MCA, and lenticulostriatal arteries, and functional organization of the region, both at the cortical and white matter levels [13,14]. First described by Penfield and Faulk as a visceral somatic region [15], it is now known that the insula is involved in several areas: sensory–motor, chemical–sensory, social–emotional, and cognitive [16].

To facilitate surgical access to the insula, several classifications have been proposed, but their predictive value remains unclear [17,18]. In particular, three variants have been described regarding the relationship of the tumor to the lenticulostriatal arteries, in order to emphasize the clinical importance of their identification, since their damage results in stroke of the internal capsule and hemiparesis [19,20]. Based on the invasiveness pattern, the YC distinguishes whether a tumor is confined to the insula (type 3A) or extends into the adjacent operculum (type 3B) and whether the insular lesions involve an orbitofrontal or a temporopolar paralimbic area (type 5A) or both (type 5B) [21]. Anatomically, according to the BSC, insular gliomas are identified according to their location above or below the Sylvian fissure and anterior or posterior to the foramen of Monro in four zones; each insular zone has a different degree of surgical accessibility and functional and vascular involvement, so the type of surgical approach is often decided according to this. Mandonnet and Duffau’s classification describes the patterns of extension of insulo-paralimbic tumors by giving greater importance to white matter tracts, suggesting the concept of function-guided surgery [22].

The main goal of the surgical treatment of gliomas is maximum safe resection, which is best expressed by the concept of “oncofunctional balance”. [23] Contributing to such surgical nihilism is the dubious benefit of resection over a safer strategy such as biopsy and adjuvant therapy. However, recent studies have shown that, even for insular lesions, surgical resection is the main prognostic factor for PFS, slowing malignant transformation and controlling comitial seizures. Key factors in achieving complete resection in this area are, therefore, determined by the knowledge of surgical anatomy and connectomics. Advances in neuroimaging and intraoperative techniques, such as electrophysiological monitoring and functional, cortical, and subcortical mapping, contribute to excellent results [13,24,25,26,27,28].

There is no general consensus on the choice of the appropriate surgical approach. Several studies have been reported in the literature regarding this, especially on the two main corridors: transcortical and transsylvian [29]. Numerous publications have been shared regarding the anatomical difficulty, but less attention has been given to the functional component of the insular approach [30]. Due to the development of intraoperative mapping, more and more surgeons have preferred to switch to the transopercular approach over the transsylvian [24]. Although the surgical approach (transcortical, transsylvian, combined) depends on the experience of the neurosurgeon, it has been shown in cadaveric studies that transcortical corridors offer better exposure for the resection of gliomas extending beyond the insula [31].

Several authors have compared these surgical techniques in terms of EOR, OS, and postsurgical neurological outcomes, and it was found that complete resection rates are not affected by the approach used during surgery [32]. It is still not entirely clear whether tumor localization within the insula (according to the BSC) has an impact on perioperative morbidity and quality of life, keeping in mind that different areas have relationships with different anatomical and functional structures. For instance, the posterior zones of the insula are more easily reached with the transcortical approach, due to the use of intraoperative mapping and cortical stimulation [5,33].

According to the literature [21], it is possible to identify negative prognostic factors, including putaminal [34] or paralimbic involvement and tumor grading, and positive prognostic factors, including clinical onset with comitial seizures, isocitrate dehydrogenase (IDH) mutation, greater extent of resection [14], age [35], female sex, and Karnofsky performance status [36]. The role of the distance between the tumor and the subventricular zone (SVZ), located along the margin of the caudate nucleus adjacent to the lateral ventricle, needs to be evaluated [37].

In our series, a total of 58 primary high-grade insular gliomas patients were enrolled. A poor Charlson comorbidity index (CCI) and MCA encasement negatively affected the OS (Table 3A). On the other hand, KPS and autonomous deambulation at 1 month after surgery, radiotherapy, and GTR positively affected the OS. In contrast, preoperative KPS, tumor volume, and chemotherapy were not significantly correlated with the OS.

Regarding resectability, zone I and IV in the BSC are considered positive predictors, while the involvement of the lenticulostriatal arteries is considered a negative predictor; accurate identification of the origin, path, and distribution of these vessels relative to the tumor is essential for good surgical outcomes, since it determines the patient’s candidacy for surgery [21]. However, in our series, the transcortical approach had a higher EOR and postoperative KPS when the lesion was in zone III-IV. Indeed, perioperative ischemic insults are the major risk factor for long-term and permanent morbidity in this surgery [19,20]. Furthermore, in the transsylvian corridor, opercular retraction causes edema and ischemia of the M3 branches. Ülgen et al. reported that tumors involving the hippocampus are more likely to have residual tumors postoperatively [38]; the presence of contrast-enhanced tumors has also been shown to predict a higher resection rate than that for tumors without contrast.

As described by Yasargil et al. with the transsylvian approach, reaching the insula is not as difficult, and the uninvolved opercular cortex is preserved. However, in the literature, complication rates (up to 30%) are higher in the transsylvian approach than in the transopercular approach [4,31]. Regarding the transcortical approach, Hameed et al. reported 255 consecutive cases of insular gliomas that underwent transcortical tumor resection, maximizing the EOR and postoperative outcomes [18]. Given the functional importance of the insular approach, with the widespread implementation of intraoperative mapping, the transopercular approach is becoming increasingly used. In particular, in a systematic review by Di Carlo et al., no difference was highlighted in terms of the GTR based on the type of anesthesia used (general versus awake) [32]; however, Pallud et al. [39] and Rossi et al. [40] reported that the EOR was significantly higher with awake mapping, with continuous intraoperative neuromonitoring complementary to subcortical stimulation to monitor verbal function.

In the literature, the rates of postsurgical transient deficits range from 14 to 59%, while permanent deficits range from 0 to 20% [14]. The results of our series are in line with the literature, both in terms of short and long deficits (Table 2). Zarino et al. analyzed 35 patients to investigate the presence of speech disturbances before and after surgery and the relationship of these disturbances with preoperative tumor volume and with the extent of resection after surgery, highlighting the trend toward cognitive recovery [16,41]. But EOR remains the main predictor especially in terms of OS, as also shown by the results of this series. Sanai et al. reported a median EOR of 82 and 81% in insular LGGs and HGGs [11], respectively, in 115 procedures; Duffau reported 51 insular LGGs with a median EOR of 77% [42].

In our series, the surgical approach used was transsylvian and transcortical in 11 and 47 cases, respectively. The probability of using the transsylvian approach was higher in cases with MCA encasement and partial resection. The probability of using the transcortical approach was higher in cases with BSC and YC ≥ 3. A possible explanation could be the surgeon’s preference for the transcortical approach in the case of larger tumors. The comparison between the two different approaches did not display differences in terms of neurological deficits and OS. The transcortical approach was related to the greater achievement of GTR. According to the BSC, the transcortical approach had a higher EOR and postoperative KPS when the lesion was in zone III–IV. An interesting result was the role of the BSC in the survival analysis (Figure 2A). Although this result was not statistically significant, patients with grade I or II of BSC had a higher risk of death. A possible explanation could be the more complex location of the lesions, with extension to the language network and primary motor; in these cases, it is more difficult to achieve complete resection, and there is a greater risk of postoperative neurological damage, which can impact adjuvant therapies and survival itself.

## 5. Limitations

Important limitations of this study should be highlighted: given the retrospective and monocentric nature of the analysis, it is crucial to interpret the results with caution. A multicenter study could provide greater heterogeneity in the selection of the two surgical approaches. In this analysis, there was an imbalance between the number of cases of transsylvian and transcortical approaches. The groups were not randomized, and the choice of one technique over another was influenced by the preferences of senior surgeons.

## 6. Conclusions

A more extensive resection of insular gliomas can be performed with an acceptable morbidity profile and is associated with improved OS. Both the transsylvian and transcortical approaches to the insula are linked to low morbidity. However, the transcortical approach, when combined with intraoperative mapping, is particularly advantageous for achieving a greater EOR, especially in large gliomas and tumors located along the inferior border of the Sylvian fissure (zone III–IV according to the BSC).

## Figures and Tables

**Figure 1 curroncol-32-00098-f001:**
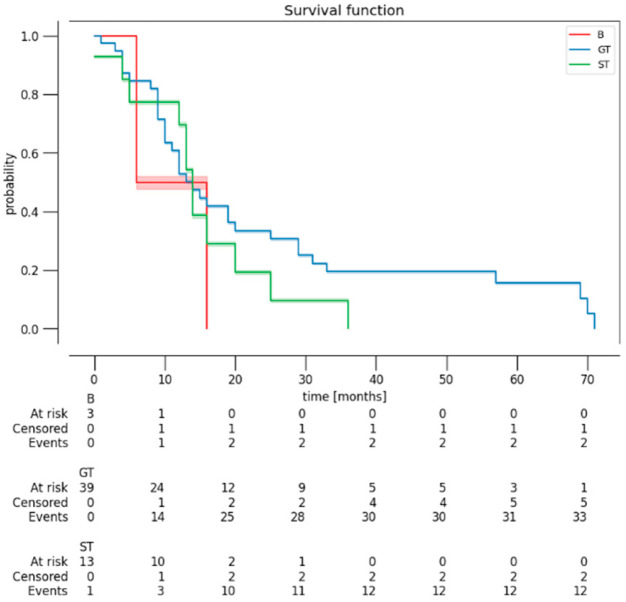
Kaplan–Meier curves revealing the OS of patients, stratified by EOR (B: biopsy; GT: gross total resection; ST: subtotal resection).

**Figure 2 curroncol-32-00098-f002:**
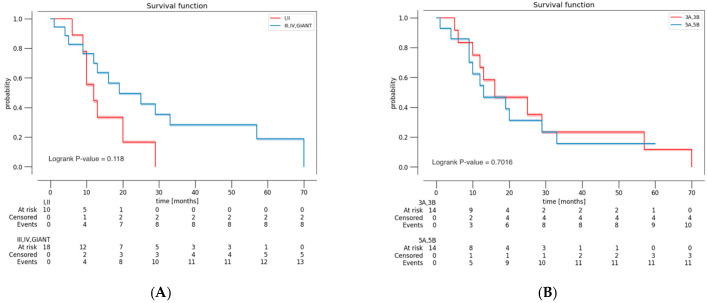
Kaplan–Meier curve demonstrating OS of patients (**A**) based on Berger–Sanai classification (<3; ≥3) and (**B**) Yaşargil classification (<3; ≥3). Berger–Sanai classification: 1 = I; 2 = II; 3 = III; 4 = IV; 5 = GIANT. Yaşargil classification: 1 = 3A; 2 = 3B; 3 = 5A; 4 = 5B.

**Figure 3 curroncol-32-00098-f003:**
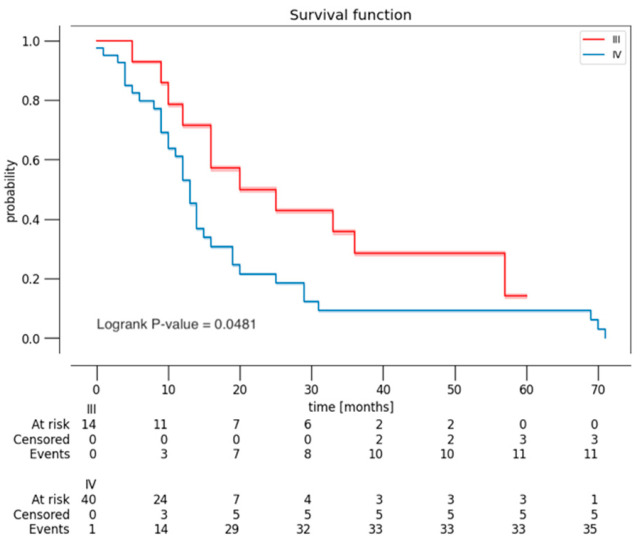
Kaplan–Meier curve demonstrating OS of patients, based on histological grade (grade 3 and 4).

**Table 1 curroncol-32-00098-t001:** Summary of patient, disease, and treatment characteristics.

Parameter	No. (%)
Age at diagnosis (yrs) Median Range	57.4 24–80
Sex M F	34 (58.6) 24 (41.4)
KPS score (median) At diagnosis Postop At 1-month follow-up	90 [range 40–100] 80 [range 40–100] 90 [range 50–100]
CCI (median)	2.9
Side of tumor Left Right	25 (43.1) 33 (56.9)
WHO tumor grade 3 4	13 (22.4) 45 (77.6)
IDH mutation	13 (22.4)
MGMT methylation	23 (36.7)
Extent of resection Total resection Subtotal resection Partial resection	40/58 (69) 15/58 (26) 3/58 (5)
Preop enhancement tumor volume (cm^3^) Mean Range	27 1.7–94.4
Preop FLAIR volume (cm^3^) Mean Range	64 1–144.5
Vascular encasement (MCA)	17 (29.3)
Surgical approach Transcortical Transsylvian	47 (81) 11 (19)
Surgical time (minutes) Transcortical Transsylvian	209.5 223.2
Home discharge	39 (67.2)
Patient postop chemotherapy Yes No	44 (75.9) 14 (24.1)
Patient postop radiation Yes No	45 (77.6) 13 (22.4)
OS (months) Median range	18.5 3–71

KPS = Karnofsky performance status; CCI = Charlson comorbidity index; MCA = middle cerebral artery; OS = overall survival.

**Table 2 curroncol-32-00098-t002:** Postoperative neurological morbidity and deficit-resolution profile.

	Outcome—Total No. of Patients (%)		
Condition	Postop	At 1-Month Follow-Up	At 3-Month Follow-Up
Motor deficit	24 (41.4)	20 (34.5)	9 (15.5) *
Speech deficit	13 (22.4)	11 (19)	9 (15.5)
Sensory deficit	17 (29.3)	14 (24.1)	/

* 4/9 patients (44.4%) had autonomous ambulation.

**Table 3 curroncol-32-00098-t003:** (**A**) Association between operative and follow-up characteristics based on OS. (**B**) Association between follow-up characteristics and surgical approach (where transcortical = 0, transsylvian = 1).

**(A)**		
**Variable**	**r**	* **p** * ** -Value**
CCI	−0.361	0.003
MCA encasement	−0.314	0.043
KPS at 1-month FU	0.409	0.001
Autonomous deambulation at 1 month	0.339	0.027
Radiotherapy	0.321	0.009
Histological	−0.261	0.027
GTR	0.231	0.045
KPS preoperative	0.160	0.122
Tumor volume	−0.224	0.146
Chemotherapy	0.007	0.479
(**B)**		
**Variable**	**r**	* **p** * ** -Value**
BSC	−0.518	0.002
YC	−0.474	0.005
MCA	0.686	0.001
Biopsy	0.284	0.015
GTR	−0.245	0.031
KPS postoperative	−0.077	0.281
KPS at 1-month FU	0.151	0.129
Autonomous deambulation at 1 month	0.006	0.486
Aphasia at 1 month	−0.130	0.231
PFS	−0.102	0.230
OS	−0.147	0.139

CCI = Charlson comorbidity index; KPS = Karnofsky performance status; MCA = middle cerebral artery; FU = follow-up; GTR = gross total resection; PFS = progression-free survival; OS = overall survival; YC = Yaşargil classification (<3; ≥3); BSC = Berger–Sanai classification (<3; ≥3).

## Data Availability

The data or information needed to re-produce the findings presented is available from the corresponding author upon reasonable request.

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
