# Peer review of "Safety and Efficacy in the Transcortical and Transsylvian Approach in Insular High-Grade Gliomas: A Comparative Series of 58 Patients"

_curroncol, 2025, doi:10.3390/curroncol32020098_

Round 1
Reviewer 1 Report
Comments and Suggestions for Authors
The manuscript " Safety and efficacy in the transcortical and transsylvian approach in insular high-grade gliomas: a comparative series of 58 patients" by Alberto Morello and colleagues contains the results of a monocentric retrospective study on a series of 58 primary high-grade insular gliomas operated between January 2019 and June 2024. The surgical approach used was transsylvian in 11 pts. and transcortical in 47. The choice between the approaches was dependent on the surgeon. Most of the differences detected by the authors between the results of the two surgical approaches did not reach statistical significance. The present study contains an in-depth discussion of the literature but, due to its retrospective and not randomized nature, it adds little to what is already known regarding insular gliomas. The discussion section consists by enlarge of a separate presentation of the pertinent literature and a limited analysis of their own result. A more integrated discussion will likely improve the manuscript. Despite these criticisms, the series is well described and the data presented may be useful for a future meta-analysis.
Minor points:
pg 10 row 5 "An interesting result was the rule of BSC..." unclear, perhaps it is role instead of rule.
pg 10 row 13 "A few limitations should be highlighted" sound reductive for a retrospective non randomized study, please rephrase.
Author Response
Thank you for pointing this out. The point of the revised manuscript is found this edit: page 10, row 5-13.
Reviewer 2 Report
Comments and Suggestions for Authors
I have read the manuscript “Safety and efficacy in the transcortical and transsylvian approach in insular high-grade gliomas: a comparative series of 58 3patients” with interest. It is well conducted study, structured accordingly to the standards of such paper, presented in appropriate English and ready to be published.
I have some minor suggestions to the authors in order to improve their paper.
The first one is related to the essence of the paper. I would recommend some discussion or conclusion regarding the indications for transcortical and for transsylvian approach- maybe related to the Berger-Sanai classification, or vessel involvement (there was a hint for that). Of course, that the outcome of both approaches will be statistically same- that is the essence of contemporary medical care, to be safe.
And second suggestion: please, the discussion of figure 3: “Patients with a lower grade of BSC have a higher risk of death. Patients with a higher grade of YC have a higher risk of death. However, these results were not statistically significant (p = 0.118 and 110.702, respectively).” Is not clear and understandable.
I wish the authors luck with future projects.
Author Response
Thank you for pointing out these issues. We agree with these comments. The places where we can find these changes in the revised manuscript are p. 6 - line 9; p. 10 - line 6; p. 10 - line 22.